# Subcutaneous Fat Necrosis and Hypercalcemia with Nephrocalcinosis in Infancy: Case Report and Review of the Literature

**DOI:** 10.3390/children8050374

**Published:** 2021-05-09

**Authors:** Katerina Chrysaidou, Georgios Sargiotis, Vasiliki Karava, Dimitrios Liasis, Victor Gourvas, Vissarios Moutsanas, Athanasios Christoforidis, Stella Stabouli

**Affiliations:** 11st Department of Pediatrics, Aristotle University of Thessaloniki, Hippokratio General Hospital Thessaloniki, 54642 Thessaloniki, Greece; cathchris84@hotmail.com (K.C.); gsargiot@yandex.com (G.S.); vasilikikarava@hotmail.fr (V.K.); christoforidis@doctors.org.uk (A.C.); 2Department of Pediatrics, St Luke’s Hospital, Panorama, 55236 Thessaloniki, Greece; dliasis@gmail.com (D.L.); vgourvas@gmail.com (V.G.); 3Pediatric Department, G. Gennimatas Hospital, 54635 Thessaloniki, Greece; aris_moutsanas@yahoo.gr

**Keywords:** subcutaneous fat, fat necrosis, hypercalcemia, newborn, nephrocalcinosis

## Abstract

Subcutaneous fat necrosis is an uncommon benign panniculitis affecting more commonly full-term newborns. It has been associated with birth asphyxia and meconium aspiration, as well as therapeutic hypothermia. Although the prognosis is generally favorable, complications such as hypercalcemia, thrombocytopenia, hypoglycemia and hypertriglyceridemia may complicate its course. The most serious complication is hypercalcemia that may reach life threatening levels and can be associated with nephrocalcinosis. We thereby describe a case of subcutaneous fat necrosis after therapeutic hypothermia, which presented with late-onset refractory severe hypercalcemia and persistent nephrocalcinosis during the follow up of the patient. Due to the risk of the development of chronic kidney disease, we highlight the importance of careful monitoring of hypercalcemia and review the literature of subcutaneous fat necrosis related to nephrocalcinosis.

## 1. Introduction

Subcutaneous fat necrosis of the newborn (SCFN) is a rare, usually self-resolving, form of panniculitis which affects more commonly full-term infants within the first weeks of life. SCFN presents as plaques or nodules, frequently erythematous, firm and dolorous, situated mainly in the extremities, shoulders, buttocks, thighs and rarely in the face. Despite generally favorable prognosis, SCFN can be complicated with hypercalcemia, hypertriglyceridemia, thrombocytopenia or hypoglycemia [1,2,3,4,5].

We present the case of an infant with SCFN, late-onset refractory severe hypercalcemia and persistent nephrocalcinosis. The aim of this paper was to highlight the rare disorder of SCFN, its potentially dangerous complication of hypercalcemia and the relation with nephrocalcinosis. Moreover, we aimed to review the literature on this rare disorder and to discuss the therapeutic challenges that arise regarding the management of calcium levels and potential effects of treatment on normal growth and kidney function.

## 2. Case Presentation

A 55-day-old female infant was referred for poor weight gain (500 g from birth), hypercalcemia and multiple painful palpable subdural nodules, without skin discoloration, located in the lateral abdomen, the face, both thighs and gluteal regions.

Immediately after birth, the neonate was admitted to the intensive care unit due to shoulder dystocia, meconium-stained amniotic fluid, meconium aspiration and perinatal asphyxia. She was born at term (38 weeks + 5 days) with a birth weight of 3850 by a 41-year-old primigravida. During pregnancy, the mother presented hypothyroidism treated with levothyroxine and gestational diabetes managed with diet. The neonate received mechanical ventilation and therapeutic hypothermia due to moderate hypoxic-ischemic encephalopathy and episodes of tonic-clonic seizures. During the last days of her hospitalization, she developed a neck abscess without any underlying palpable swelling, which was ruptured, drained and treated with antibiotic therapy. She was discharged at the age of 4 weeks without any other skin lesion and receiving no further treatment (Figure 1).

Notwithstanding the abnormal skin findings, the clinical examination at the time of her admission to our department at the age of 55 days was unremarkable without any definitive signs of congenital defects, with the exception of slight hypotonia. The complete blood count was normal; electrolyte levels, kidney and liver function were within a normal range, but with both increased total serum calcium (12.3 mg/dL) and ionized serum calcium levels (7.19 mg/dL). Triglyceride levels were also elevated (312 mg/dL), whereas parathormone and 25-OH vitamin D levels were consistently suppressed in multiple assessments (minimum values: 0.9 pg/mL and 6.7 ng/mL, respectively). Skin biopsy that was obtained from the gluteal region confirmed the diagnosis of SCFN (Figure 2).

Treatment with hyperhydration and furosemide (2 mg/kg body weight) failed to lower total calcium levels (maximum 14.23 mg/dL). As a result, intravenous methylprednisolone at a dose of 2 mg/kg was initiated, resulting in the normalization of calcium levels after 7 days of treatment. Hypercalciuria was a constant finding during treatment; the calcium/creatinine ratio ranged from 0.12 to 2, and 24 h urine calcium levels were 5.5 mg/kg/day. Gradual tapering and discontinuation of furosemide and methylprednisolone were associated with the reoccurrence of hypercalcemia on the 29th day of hospitalization (total serum calcium 12.3 mg/dL). Treatment with low-dose oral prednisolone was then reinitiated (0.5 mg/kg body weight), resulting in stable calcium levels between 10 and 11.5 mg/dL. Ultrasonographic examination revealed bilateral nephrocalcinosis, without other structural or functional anomalies of the kidneys and the urinary tract (Figure 3). Nephrocalcinosis persisted in repeated ultrasound examinations with normal kidney function. 

On the 5th and 23rd day of hospitalization, the patient presented 2 episodes of febrile urinary tract infections. At the 1st urinary tract infection, urine culture was positive for Escherichia coli, resistant to β-lactamases and at the 2nd for Pseudomonas aeruginosa. Voiding cystography performed 15 days later showed grade III vesicoureteral reflux in the right ureter.

At the age of 5 months, the patient remained asymptomatic with a gradual improvement of the skin lesions. She continued treatment with oral prednisolone treatment (0.4 mg/kg/day) with regular follow up in the ambulatory day care unit. Total serum calcium remained within the normal range, but follow-up ultrasonographic assessment of the kidneys showed persistent nephrocalcinosis.

## 3. Discussion

Subcutaneous fat necrosis of the newborn is an uncommon benign disorder affecting more commonly full-term infants, within the first weeks of life, characterized by firm, palpable subcutaneous nodules or plaques with or without erythema. Lesions may appear isolated or clustered typically on the shoulders, back, buttocks and face [6]. Despite its self -limiting nature, significant complications may occur, such as hypercalcemia, hypoglycemia, hypertriglyceridemia and thrombocytopenia. 

Numerous neonatal and maternal conditions have been associated with SCFN, including perinatal asphyxia; therapeutic hypothermia; meconium aspiration; obstetric trauma; sepsis; and maternal diseases, such as preeclampsia and diabetes [1,2,3]. While the pathogenesis remains unclear, several hypotheses have been proposed. Neonatal distress could affect the normal blood supply to the neonatal fat tissue, leading to necrosis in an attempt to shunt blood from skin and adipose tissue to vital organs, the heart and the brain [7]. The composition of neonatal fat with increased saturated fatty acids and its higher melting point compared to adult fat have been associated with the tendency for solidification and crystallization [8]. Therapeutic hypothermia has been suggested to play an important role, as the higher ratio of saturated to unsaturated fat in the skin of neonates predisposes to hypothermia-induced crystallization [9,10]. Therapeutic hypothermia offers neuroprotection in infants ≥35 weeks of gestation with moderate to severe hypoxic ischaemic encephalopathy between 1 and 6 h of life aiming to lower the temperature of the vulnerable deep brain structures to 33–34 °C and to reduce cerebral metabolic rate [11]. A series of 30 infants with SCFN over a 20-year period reported that 90% of the infants had a history of complicated delivery or perinatal stress and 50% had maternal comorbidities (38% diabetes mellitus, 25% gestational diabetes, 6% preeclampsia, 6% hypothyroidism, 6% placenta abruption and 19% other). Hypoxic ischemic encephalopathy was reported in 40% of the patients with SCFN, while 91% of them were treated with therapeutic hypothermia [1]. According to a recent metanalysis of 126 SCFN cases, 86% of the patients experienced intrauterine and/or perinatal distress or pregnancy complications, and a significant percentage of neonates were delivered via caesarian section due to fetal distress. Moreover, in 20% of the cases, the patients were treated with therapeutic hypothermia [8]. However, the incidence of SCFN in cooled neonates with hypoxic ischaemic encephalopathy in the literature is lower, ranging from 3 to 8% [12,13,14,15]. Whether SCFN is a direct complication of hypoxia/ischaemia or develops secondarily due to the exposure to cold temperature or the combination of both requires further investigation. 

Histopathologically, SCFN is characterized by granulomatous reaction, lobular panniculitis and multinucleated giant cells with crystals [16]. The changes are more prominent at the interface between the dermis and subcutaneous fat, with inflammatory infiltrate extending from the perivascular area to the necrotic adipocyte tissue. SCFN is characterized by needle-shaped clefts in histiocytes and lipocytes, and inflammation reaching to the depths of subcutaneous tissue [17,18]. The most important differential diagnosis for SCFN is sclerema neonatorum, which is characterized by diffuse hardening of the skin and subcutaneous fat that becomes bound down and adheres to underlying muscle and bone adversely affecting basic functions, such as breathing, feeding and movement. Mortality is high, and it mainly affects critically ill, premature infants with low birth weight [19]. Differential diagnosis also includes erythema nodosum, bacterial cellulitis, histiocytosis, Farber’s disease or fibromatosis and rhabdomyosarcomas [20].

SCFN is usually reported to be benign and painless, although there is a published case that presented with severe pain and required opiate analgesia [21]. Although lesions improve spontaneously within a few weeks to months, SCFN has been associated with metabolic complications, such as hypoglycemia; hypertriglyceridemia; or with hematological complications, such as anemia and thrombocytopenia [4]. All complications except hypercalcemia are self-limiting and respond to treatment. Thrombocytopenia appears just before or simultaneously with skin lesions and disappears without bleeding issues with the resolution of lesions. It can be caused by the sequestration of platelets within the subcutaneous tissue, whereas increased prostaglandin synthesis in the necrotic area was suggested too. Another complications is hypoglycemia, which may be associated with perinatal distress and maternal gestational diabetes and hypertriglyceridemia, as a cause of lipid release resulting from the necrosis of fat tissue [4]. SCFN is a rare cause of hypercalcemia in the neonate. Hypercalcemia develops when skin lesions begin to resolve, usually between one to six months after the development of subcutaneous nodules [2] and occurs more often secondary to asphyxia [22]. The mechanism which is responsible for this complication is not fully elucidated. Subcutaneous granulomas secrete 1,25-(OH)_2_ vitamin D, which stimulates the intestinal reuptake of calcium, resulting in hypercalcemia. High concentration 1-alpha-hydroxylase-produced activated macrophages are considered to be responsible for this increase [23]. Other investigators suggested that bone resorption occurs due to elevated parathyroid hormone (PTH) and prostaglandin E2 levels, whereas it has also been hypothesized that calcium is mobilized from resolving areas of subcutaneous fat necrosis [23].

In the aforementioned metanalysis of 126 patients, 51% had hypercalcemia, and in 57% of them, hypercalcemia presented in the first 28 days of life, whereas in 30% after the first 28 days of life. Most of the patients (77%) developed hypercalcemia within 30 days of skin lesion onset and 95% within 60 days [8]. Del Pozzo-Magana et al. reported hypercalcemia in 63% of the patients [1] in accordance with the previously reported incidence in other case series (29–69%) [2,22]. A lower incidence has been also reported (25%) [24,25]. Hypercalcemia can lead to gastrointestinal disorders, and neurological and cardiac life-threatening complications. Hypercalcemia may reduce renal concentrating capacity presenting with polyuria and dehydration leading to kidney failure and metastatic calcifications [4,5,8,26,27]. Metastatic calcifications have been reported to be rarely found in kidneys, skin, myocardium, liver, inferior vena cava, falx cerebri and gastric mucosa [4,20,27]. Symptomatic patients were reported to exhibit polyuria, polydipsia, irritability, seizures, constipation, hypotonia, failure to thrive and persistent nephrocalcinosis [8]. Nephrocalcinosis can be transient or persistent, affecting renal function; the diagnosis is often made by renal sonography and can be confirmed by computerized tomography (CT) [28]. In a cohort of patients with severe hypercalcemia, nephrocalcinosis was present in 83% of them, suggesting the association of the severity of hypercalcemia with nephrocalcinosis [5]. Generally, SCFN is a self-healing condition, but hypercalcemia necessitates aggressive treatment to safeguard kidney function and bone health.

The mainstay of treatment includes hyperhydration, intravenous loop diuretics (furosemide) and systemic corticosteroids [3]. A low-calcium and -vitamin D diet could be a therapeutic option, but there are significant concerns of using low-calcium and -vitamin D formulas due to the potential impact on normal growth during infancy. Moreover, treatment with corticosteroids (dexamethasone and hydrocortisone) in the neonatal period has been associated with the impairment of somatic growth during treatment (“early growth retardation”) and during follow up at ages ranging between 2 and 17 years [29,30,31,32]. Both furosemide and corticosteroids cause hypercalciuria and thus, increase the risk of nephrocalcinosis [12]. Bisphosphonates and calcitonin, which decrease bone calcium reabsorption, have been used in refractory cases. Corticosteroids act slowly in comparison with pamidronate, achieving normalization in serum calcium within a week. They act by interfering in the metabolism of vitamin D and increasing renal calcium excretion [33]. Pamidronate, which belongs to the bisphosphonates group, inhibits the activity of osteoclasts and blocks the dissolution of bone calcium phosphate crystals [34]. Pamidronate has been used as first-line treatment in case series in order to avoid nephrocalcinosis, and despite the avoidance of furosemide and glucocorticoid therapy, nephrocalcinosis was observed, associated with greater disease severity and duration of hypercalcemia [35,36,37] The prognosis of hypercalcemia and nephrocalcinosis varies in the published literature. It has been reported that in 76% of the cases, hypercalcemia was resolved within 4 weeks, and in 88% of the cases, up to 84 days of life. However, one patient was reported to present persistent hypercalcemia up to 1 year of age [8]. It has been suggested that screening for hypercalcemia should be continued up to 6 months after the resolution of skin lesions [33,38,39,40]. Nephrocalcinosis is resolved in most cases [2], but in a previously mentioned series of seven cases of severe hypercalcemia, it persisted for up to 4 years of follow up [5]. 

In Table 1, published cases of SCFN complicated with nephrocalcinosis are presented.

## 4. Conclusions

In conclusion, SCFN is a rare disorder which can lead to serious complications that require close follow up. Since the use of therapeutic hypothermia in neonates is increasing, SCFN and hypercalcemia should be suspected by pediatricians and neonatologists in infants with clinical and laboratory characteristic manifestations. Hypercalcemia is a serious complication that should be identified and treated early in order to prevent the development of nephrocalcinosis.

## Figures and Tables

**Figure 1 children-08-00374-f001:**
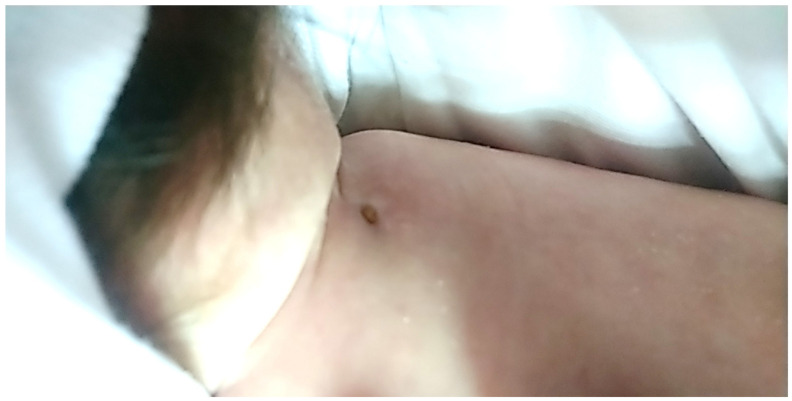
Neck abscess during healing.

**Figure 2 children-08-00374-f002:**
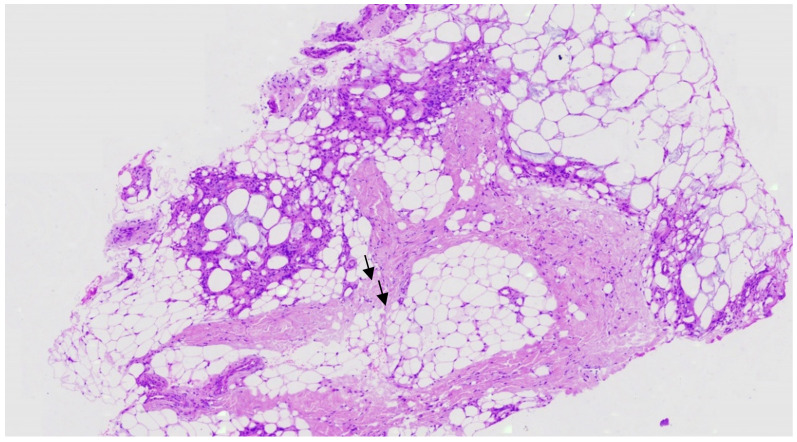
Normal epidermis from the gluteal region with granulomatous inflammation, panniculitis and basophilic fat necrosis with intracytoplasmic needle-shaped clefts in histiocytes and lipocytes (arrows).

**Figure 3 children-08-00374-f003:**
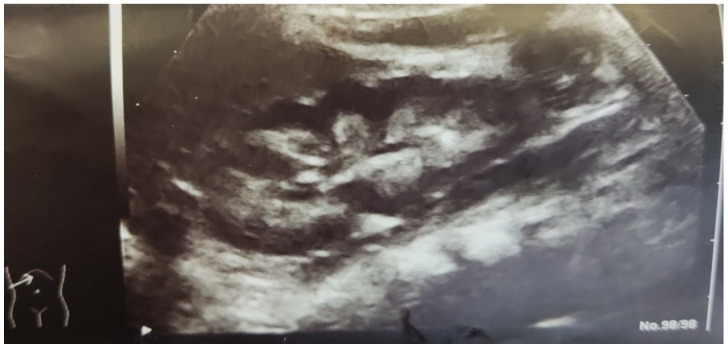
Ultrasonography of the kidney showing increased echogenicity of the medullary pyramids (nephrocalcinosis).

**Table 1 children-08-00374-t001:** Summary of SCFN cases complicated with nephrocalcinosis and nephrolithiasis.

Study	Case	Age	Calcium Level	Treatment	Outcome
Mahé, E. et al. Br J Dermatol, 2007 [2]	3/16 patients with SCFN with nephrocalcinosis	infant period	most severe hypercalcemia > 3 mmol/L	Patient 1: furosemide+ prednisone Patient 2: furosemide+ pamidronate Patient 3: furosemide alone In all hydration	1/3: transient renal insufficiency in all (3/3) complete resolution of hypercalcemia, nephrocalcinosis and renal insufficiency
Khan, N., A. Licata, and D. Rogers, Clin Pediatr (Phila), 2001 [41]	1 patient with nephrocalcinosis	7-week-old boy	4.35 mmol/L	hydration, furosemide, prednisone, calcium free formula	at age 3 months, persistent stable bilateral nephrocalcinosis -normal renal function
Gu, L.L. et al., Pediatr Radiol, 1995 [28]	2 patients with nephrocalcinosis and nephrolithiasis	patient 1: 6-week-old boy patient 2: 2^1/^2-month-old infant	3.87 mmol/L 3.63 mmol/L	not available	1: resolution of nephrolithiasis at the age of 7 months and nephrocalcinosis at the age of 15 months 2: resolution of nephrolithiasis and some decrease in the degree of increased echogenicity of the renal cortex at follow up at 7 and 15 months
Tran, J.T. and A.P. Sheth, Pediatr Dermatol, 2003 [4]	1 patient with nephrolithiasis	1-month-old boy	18.4 mg/dL	hydration, furosemide, polycitrate, low calcium and vitamin D free formula	not available
Canpolat, N. et al., Turk J Pediatr, 2012 [7]	1 patient with nephrocalcinosis	4-month-old girl	13.5 mg/dL	hydration, discontinuation of vitamin D, low-calcium diet, potassium citrate	persistence of bilateral grade III nephrocalcinosis until the age of 5 years, with normal renal function
Alos, N. et al. Horm Res, 2006 [36]	4 patients with nephrocalcinosis	patient 1: 42-day-old boy patient 2: 30-day-old girl patient 3: 13-day-old girl patient 4: 20-day-old girl	2.19 mmol/L 1.58 mmol/L 1.64 mmol/L 1.49 mmol/L	hydration, furosemide, diet low in calcium and vitamin D, pamidronate in patients 1,2,3 without furosemide in patient 4	Resolution of nephrocalcinosis Patient 1: at the age of 3 years Patient 2: 7 months Patient 3: 2 months Patient 4: 3 months
Shumer, D.E. et al., Arch Dis Child Fetal Neonatal Ed, 2014 [5]	5/7 patients with severe hypercalcemia due to SCFN, who developed nephrocalcinosis	patient 1: 38-day-old boy patient 2: 32-day-old male patient 3: 35-day-old girl patient 4: 21-day-old boy patient 5: 28-day-old girl	4.4 mmol/L 5.1 mmol/L 4.1 mmol/L 3.8 mmol/L 4.8 mmol/L	1: hydration, furosemide, glycocorticoid, dietary calcium restriction 2: the same with patient 1 plus calcitonin and citrate 3: hydration, furosemide, glycocorticoid 4: hydration, furosemide, glycocorticoid, pamidronate, citrate, dietary calcium restriction 5: hydration, furosemide, glycocorticoid, citrate, dietary calcium restriction	persistent on the most recent ultrasound 1: 10 months 2: 20 months 3: 8 months 4: 48 months 5: 36 months
Stefanko, N.S. and B.A. Drolet, Pediatr Dermatol, 2019 [8]	1 patient with nephrocalcinosis	41-day-old girl	17.2 mg/dL	furosemide, calcitonin, prednisone, bisphosphonates	persistent nephrocalcinosis until the age of 8 months
Del Pozzo-Magaña, B.R. and N. Ho, Pediatr Dermatol, 2016 [1]	3/30 cases of SCFN with nephrocalcinosis	18 months after the diagnosis of SCFN	not available	not available	persistent after 2 years despite treatment
Tuddenham, E., A. Kumar, and A. Tarn, BMJ Case Rep, 2015 [42]	1 patient with nephrocalcinosis	1-month-old girl	4.34 mmol/L	hydration, prednisone, furosemide,	nephrocalcinosis remains until the age of 2 years
Vijayakumar, M. et al., Indian Pediatr, 2006 [43]	1 patient with nephrocalcinosis	1-month-old infant	17.8 mg/dL	hydration, furosemide, etidronate, potassium citrate	nephrocalcinosis until the 2nd month, then data not available
Aucharaz, K.S. et al., Horm Res, 2007 [44]	1 patient with nephrocalcinosis	1-month-old girl	3.99 mmol/L	hydration, furosemide, prednisone and low calcium formula milk	nephrocalcinosis until the 9th month, then data not available
Tizki, S. et al., Arch Pediatr, 2013 [45]	1 patient with nephrocalcinosis	1-month-old infant	3.9 mmol/L	hydration, furosemide, corticosteroids	nephrocalcinosis until 6th month, then data not available
Trullemans, B., J. Bottu, and J.P. Van Nieuwenhuyse Arch Pediatr, 2007 [46]	1 patient with nephrocalcinosis	1-month-old infant	15.1 mg/dL	hydration, furosemide, prednisone, etidronate, low calcium and vitamin D free formula	nephrocalcinosis until the age of three years old
Borgia, F. et al., J Paediatr Child Health, 2006 [47]	1 patient with nephrolithiasis	1-month-old boy	16.2 mg/dL	hydration, furosemide, prednisone, low-calcium and vitamin D- free formula	not available
Nair, S. et al. [48]	1 patient with nephrocalcinosis	1-month-old boy	18.8 mg/dL	hydration, furosemide, prednisone, alendronate, potassium citrate	significant reduction in nephrocalcinosis at 3 months of age
N, O.B. and B. Hayes BMJ Case Rep, 2019, [49]	1 patient with nephrocalcinosis	1-month-old girl	3.11 mmol/L	hydration, furosemide, low calcium formula	resolution by 4 months
Mitra, S., J. Dove, and S.K. Somisetty, Eur J Pediatr, 2011 [50]	1 patient with nephrocalcinosis	5-day-old boy	3.12 mmol/L	hydration, furosemide, prednisone	not available

## Data Availability

Data of the study can requested from the author.

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
