# Peer review of "Subcutaneous Fat Necrosis and Hypercalcemia with Nephrocalcinosis in Infancy: Case Report and Review of the Literature"

_children, 2021, doi:10.3390/children8050374_

Round 1
Reviewer 1 Report
Comments
Page 2 line 56. The main text is not clear when the total (12.3 mg/dl) and serum calcium (7.2 mg/dl) levels were measured? Is it before patient discharge (4 weeks)? What was the timeframe of antibiotic treatment?
Page 3 line 77- Please indicate clearly when the treatment was reinitiated with low dose of oral prednisone?
Page 4 line 119-For education, perhaps and to help alert the broader population about the medical terminology, can authors briefly describe “body cooling”?
figure 2- It will be more informative if different markers (arrow/asterisk/dots) represent the site of inflammation, fat necrosis, lipocytes, and histiocytes in the histology image.
Minor comments
Page 1 line 27-Please correct typo-“Plague” to “plaque.”
Page 4 line 109-Please correct typo-“Placenta abruptio” to “Placenta abruption.”
Reviewer 2 Report
In the present manuscript, the authors discussed a case study of infant subcutaneous fat necrosis. The author clearly discussed the case’s symptom, development process, and treatment method. Based on literature review, the authors concluded the potential disease and problems caused by the subcutaneous fat necrosis in clinical field.
This manuscript contained case report and literature review which are important to current subcutaneous fat necrosis diagnosis and therapy. However, this manuscript has serious language and grammatical problem. I could suggest the author carefully address this problem and revise the manuscript.
Major comments:
- For current therapeutic method in subcutaneous fat necrosis, the author listed serval cases in Table 1. Besides this table, I would suggest the author conclude the most common treatment method in the discussion or conclusion part.
- There are some grammatical and punctuation error. Some sentences are hardly to be understand. Please carefully check the language and revise it accordingly.
Minor comments:
- Please indicate the sample source or location in Figure 2 legend. And please double check the language and grammar in this legend.
- Please indicate the examination method in Figure 3 legend.
- The author mentioned “While the pathogenesis remains unclear, several hypotheses have been proposed” (line 98). Please briefly describe the hypothesized pathogenesis which have been discussed in current or most recently publications.
